# Effects of Short-Term Uncertainties on the Revenue Estimation of PPP Sewage Treatment Projects

**Qian Liu [1,*], Zaiyi Liao [1,2,*], Qi Guo [1], Dagmawi Mulugeta Degefu [2] , Song Wang [3] and Feihong Jian [1]**

[1] College of Hydraulic and Environmental Engineering, China Three Gorges University, Yichang 443002, China; guoqi@ctgu.edu.cn (Q.G.); 201808150011001@ctgu.edu.cn (F.J.)

[2] Department of Architectural Science, Ryerson University, Toronto, ON M5B2K3, Canada; dagmawi.degefu@ryerson.ca

[3] School of Information and Engineering, Zhengzhou University, Zhengzhou 450001, China; wangsong61@163.com

[*] Correspondence: 2015210101003@ctgu.edu.cn (Q.L.); zliao@ryerson.ca (Z.L.); Tel.: +86-13545837405 (Q.L.); +1-416-979-5000 (Z.L.)

**Abstract:** Many sewage treatment facilities in China have been developed and operated using the Public-Private Partnerships (PPP) model. However, a big challenge faced by these PPP projects is that the subsidy requested from the government during the operation is normally higher than what was estimated originally, thus often exceeding the budget that the government can afford. This leads to a high risk of project failure or insufficient operation. This problem is largely caused by the uncertainties exhibited in the existing models, such as the Black-Scholes (BS) model. While being used in the budget estimation, these models cannot sufficiently account for the short-term uncertainty that may be incurred during the operation of projects. In this study, a method to account for this short-term uncertainty has been proposed to improve the BS model. This allows for investigations to address issues related to how the real option value of a government subsidy is affected. A Monte Carlo simulation was performed to take into account the noise in the estimation. A sensitivity analysis further revealed that this discrepancy is largely affected by the values of the relevant parameters in the short-term uncertainty model. We found that the short-term uncertainty has a significant effect on private revenue and government subsidy and that the changes of the latter are more sensitive to the change of short-term uncertainty. The value of the relevant variables of short-term uncertainty determined the fluctuation of the revenue. The mean value and revenue had a positive correlation. The reverting speed and revenue showed a negative correlation. The short-term volatility had a positive correlation toward the fluctuation range of the revenue. The simulation results indicate that this enhanced method can produce more accurate information for a better assessment of the PPP project under a wide range of uncertainty scenarios, allowing for the best decision making by the government.

**Keywords:** public-private partnerships; real options; short-term uncertainty; mean reversion; decision-making

## 1. Introduction

The rapid industrialization and urbanization in China have significantly increased the discharge of urban sewage, which requires a large number of sewage treatment facilities [1,2]. Due to limited public funds, Public-Private Partnerships (PPP) has been adopted in the development of these urgently needed facilities, encouraging investments from potential private resources [3]. This approach has also

obtained large popularity at the international level. There are advantages for both the government and the private sector. First, this partnership allows the government to mitigate their financial burden, which would be exacerbated by large up-front investments, and it allows the government to pay for the usage of the project over time, but not immediately [4]. Second, it can shorten the time needed to provide the public with the facilities needed by taking advantage of relatively higher operation efficiency in the private sector. Third, the PPP model allows the government to focus on their tasks by transferring this infrastructure construction demand to private sectors.

In order to attract the private sector to participate in these unprofitable infrastructure projects, the government always provides various forms of incentives such as subsidies to partially offset the cost, tax relaxing policies, land development opportunities, etc. [5].

However, PPP arrangements also have many implementation problems. The risk allocation from the public sector to the private sector leads to a higher financing cost of PPP arrangements [6]. The long concession period and the complexity of most PPP projects could result in the unsuccessful implementation of the project. In addition, uncertain events make it impossible to forecast the outcomes of a PPP project accurately. Hence there is a possibility that projects being conducted using this arrangement could lead to overspending of the government budget, thereby ending up being public debt [4,7].

As discussed and concluded in many studies, there exist critical risk or uncertainty factors that may adversely influence the actual performance of urban sewage treatment PPP projects [2,8,9].

Questionnaire surveys conducted by Ke and Ameyaw concluded that the uncertainties in a typical PPP project can be classified into the following categories: political, macro-economic, operational, business, land and construction, and force majeure [10,11]. Xu et al. [12] also stated that the performance of PPP water projects was influenced by contract conditions, legislations, concession prices, inaccurate market forecasts, financing, policies and market demand changes, the macro economy, government credits and technical uncertainties. Li et al. [13] pointed out that uncertainties associated with PPP projects can be classified into macro, medium, or micro level uncertainties, each level incorporating several sub-uncertainties. Meanwhile, Shen et al. [14] classified PPP projects' uncertainties into internal risk, external risk, and project risk. Hence, it can be said that there is no standard way to classify the risks associated with PPP projects.

In this paper, uncertainties are classified into long-term and short-term uncertainties in order to study the influence of time duration on project revenue and government subsidy [15]. Long-term uncertainties are the uncertainties leading to the long-term trend of volatility of economic indicators, and short-term uncertainties are the uncertainties leading to the short-term volatility of economic indicators. Long-term uncertainties include the supply-demand structure, macro economy, macro-policy, etc. Short-term uncertainties include unexpected events, such as short-term financial crises and political instability, natural calamities, etc.

These multiple uncertainty factors make it difficult to ex-ante forecast the treatment capacity (volume) and revenue precisely [16]. The risks and uncertainties associated with the external environment in PPP projects, such as the economic, political, financial factors, and the demand conditions, are not under the direct control and influence of either party. Therefore, in an attempt to resolve this uncertainty problem in the value forecasting, more and more researchers have studied the use of Real Option Theory (ROT) [17]. ROT was first proposed by Black and Scholes, and Merton, as a pricing method in finance, and later it was introduced to assess the value of real assets [18,19]. Cheah and Liu proposed ROT to calculate the government subsidy by using a Monte Carlo simulation [20]. Carbonara developed a ROT-based model to fairly set a minimum threshold of revenue subsidy for both the private sector and government [16].

However, these studies have also demonstrated that the value deviation in the revenue prediction by using ROT cannot be explained. Some researchers pointed out that the basic assumption of ROT considers the continuous changes (long-term uncertainty) as a Geometric Brownian Motion (GBM) [21,22]. However, the occurrence of some unexpected uncertainties (short-term uncertainty)

complicates the movement of the asset value and may have a great influence on the performance and sustainability of the PPP project [23].

Many papers have concluded that the short-term uncertainty will lead to a value estimation error due to the deviation from the prediction value of the Black-Scholes (BS) model [24–26]. According to Bonis, the prediction error is influenced by both the long-term component and short-term component [27]. However, Bonis only considers a zero mean value noise, which does not accord with the practice. Posen's study indicated that the short-term uncertainty or noise may lead to a negative effect on the real option value, which is opposite to the effect of the long-term uncertainty [28].

A mixed method was proposed to describe the compound process of long-term geometric Brownian and short-term Poisson jumps [29]. However, the movement tendency of the short-term uncertainty cannot be interpreted by Poisson jumps. Childs developed a Noise Real Assets Theory (NRAT) to study the effect of short-term uncertainty which follows a mean-reverting process to the value of real assets [26].

A limitation from the previous studies is that, although they gave their identified uncertainty factors and demonstrated the existence of a short-term uncertainty, they fail to evaluate long-term and short-term uncertainties in one statistical model and clearly explain the reasons why the theoretical value of the PPP subsidy deviate from the real value. Is this deviation caused by a short-term uncertainty? If it is, how does a short-term uncertainty influence the real option value? The current study aims to address these research questions.

With recognition of the drawbacks of previous research, the authors of this paper have studied an improved mathematical method for real options with an aim to narrow the gap between the theoretical result and the practice and studied the relationship between short-term uncertainty and real option revenue. To do so, a method to model the short-term uncertainty has been introduced into the BS model, which has then been compared with the traditional approach.

This article is organized as follows: In Section 1, the theoretical basis of the BS model is summarized to illustrate the effect of short-term uncertainty. Section 2 details a collar revenue subsidy model for the private sector where the underlying asset revenue is developed. Section 3 presents how the short-term uncertainty asset model is applied to the framework of PPP projects. The influence of short-term uncertainty on the real option value is studied and analyzed. Finally, Section 4 summarizes the numerical results and draws the major conclusions.

## 2. Methodology

### 2.1. Sewage Revenue in Practice

The revenue of a typical PPP project is a combined result of the MRG (minimum revenue subsidy) and ERS (excess revenue sharing mechanisms) [30,31]. This mechanism balances the profit of projects by providing the private sector support against revenue risks borne from low revenue and giving the government the right to capture the excess revenue [32]. When the revenue fluctuates wildly, the MRG and ERS mechanisms can improve welfare by sharing the risks. In practice, the two parties will negotiate to determine terms that are mutually agreeable.

As shown in Figure 1, if the sewage treatment revenue $X_t$ in the year t is below the lower threshold $X_-$, the government provides a subsidy to cover the shortfall. The subsidy given to the private sector will then be $X_- - X_t$. If the sewage treatment revenue in the year t $X_t$ is higher than the upper threshold $X_+$, the government will gain the excess revenue $X_t - X_+$. The mathematical expression of the effective revenue for the private sector in the year t is described in Equation (1) [33].

$$X(t) = \min[\max(X_t, X_-), X_+] \tag{1}$$

where, $X_- = \theta_{min} X_*$ $X_+ = \theta_{min} X_*$, $(\theta_{min}, \theta_{max})$ is the level of subsidy, p is the unit price of the sewage treatment, and $X_*$ is the benchmark revenue.

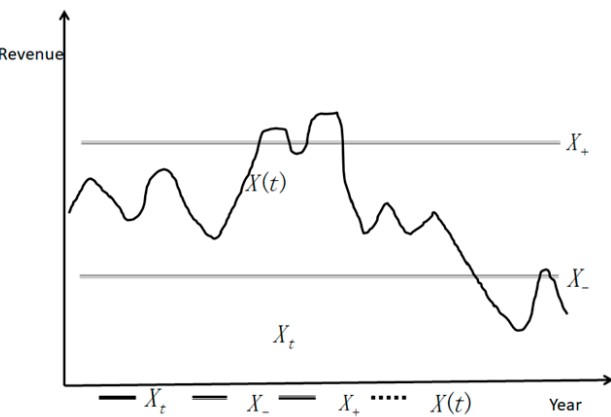

**Figure 1.** The Collar option mechanisms.

The revenue in year t can be computed as follows:

$$\text{NPV} = \sum_{t=1}^{T_o} \frac{\min[\max(X_t, X_-), X_+]}{(1+r)^{t+T_c}} - \sum_{t}^{T_o} \frac{C_t}{(1+r)^{t+T_c}} - \sum_{i=1}^{T_c} \frac{I_i}{(1+r)^i} \tag{2}$$

where C is the operation and maintain costs; I is the Construction cost; $T_c$ is the construction period; $T_o$ is the operation period; and t is the $t^{th}$ year of the operation period. r is the benchmark discount rate of industry.

Similarly, the government subsidy in the year t can be expressed below:

$$S(t) = X_t - X(t) = X_t - \min[\max(X_t, X_-), X_+] \tag{3}$$

The government subsidy can be considered as a set of independent European options which should be exercised at the end of the year. The gross value of the government subsidy over the whole period $T_c + T_o$ can be calculated in Equation (4):

$$S = \sum_{t=1}^{T_o} \frac{S(t)}{(1+r)^{t+T_c}} = \sum_{t=1}^{T_o} \frac{X_t - \min[\max(X_t, X_-), X_+]}{(1+r)^{t+T_c}} \tag{4}$$

where $T_c$ is the construction period and $T_o$ is the operation period. t is the $t^{th}$ year of the operation period.

The most popular way to simulate the motion of real asset value under long-term uncertainty is to use a BS model [33,34].

Let $X_t$ represent the theoretical value on the condition of a complete and efficient market without noise. It is assumed that the value of the asset $X_t$ follows a lognormal stochastic process. It can be described mathematically as follows [35]:

$$dX_t = \mu_X X_t dt + \sigma_X X_t dW_X \tag{5}$$

where $dX_t$ is the increment in revenue during a short period of time dt, $\mu_X$ is the revenue growth rate in a short period of time dt, and $\sigma_X$ is the long-term volatility of the annual growth rate of revenue. $\mu_X$ and $\sigma_X$ are constants, $dW_X$ is independent and identically distributed(i.i.d.) the incremental quantity from a standardized Wiener process.

For convenience, we can transform Equation (5) into a normally distributed process. The evolution of the revenue can be modeled as a function of the previous value, as shown in Equation (6):

$$dX_t = \left(\mu_X - \frac{1}{2}\sigma_X^2\right)dt + \sigma_X dW_X \tag{6}$$

The real sewage treatment revenue in the $(t + 1)^{th}$ year has a strong correlation with that of the $t^{th}$ year. This stochastic evolution process can be expressed in Equation (7):

$$X_{t+1} = X_t e^{(\mu_X - \frac{1}{2} \sigma_X^2) \Delta t + \sigma_X \varepsilon \Delta t} \qquad (7)$$

This suggests that the logarithm of revenue follows a normal distribution with a mean $(\mu_X - \frac{1}{2} \sigma_X^2) d_t$ and variance $\sigma_X^2 d_t$.

## 2.2. Short-Term Uncertainty and Ornstein-Uhlenbeck Process

It is assumed in the BS model that the market is efficient and that the information is accurate. However, the practice does not conform with the assumption. This is because the real project value may be influenced by the change of the market, economy, and policy in a short period. If the market is incomplete, meaning that the decision maker cannot know all of the future information of the market, the real option value is bound to a short-term uncertainty. We consider this short-term uncertainty to be noise. Fischer Black points out that this noise renders observations of the asset imperfect [18].

As a real asset, the sewage treatment PPP project is subjected to a lot of short-term uncertainties which cover up the real value. These short-term uncertainties are caused by exogenous factors such as unexpected policy change and economic depression, which indicate an incomplete real asset market. These short-term uncertainties lead to errors in the value prediction of real options, and the constantly changing market condition will cause the persistence of noise. The market data indicates that short-term uncertainty exhibits mean-reverting behavior [36,37]. We model the value caused by short-term uncertainty $Y_t$ as an Ornstein-Uhlenbeck (O-U) process [38], based on the assumption that the real revenue of sewage treatment will not deviate from the estimated revenue for a long time as a result of the regulation of the government.

$$dY_t = k(u - Y_t)dt + \sigma_Y dW_Y \qquad (8)$$

where k is the mean-revert speed of the short-term uncertainty. u is the mean revenue of the short-term uncertainty, $\sigma_Y$ is the volatility of the short-term uncertainty, and $dW_Y$ is i.i.d. the increment from a standardized Wiener process.

The solution to this O-U process can be expressed as:

$$Y_{t+1} - Y_t = k(u - Y_t)\Delta t + \sigma_Y dW_Y \qquad (9)$$

We can use the mean value of the annual noise of historical data as the expected mean value of short-term uncertainty. We suppose the real revenue follows a compound process, comprising a Geometric-Brownian motion and an O-U process. This dynamic process can be written as below.

$$Z_t = X_t + Y_t \qquad (10)$$

$$dz_t = \left(\mu_X + ku - kY_t - \frac{1}{2} \sigma_X^2\right)dt + \sigma_X dW_X + \sigma_Y dW_Y \qquad (11)$$

In order to take into account the noise in this model, we adopt a Monte Carlo simulation [39,40]. The conceptual framework of the model is exhibited in Figure 2.

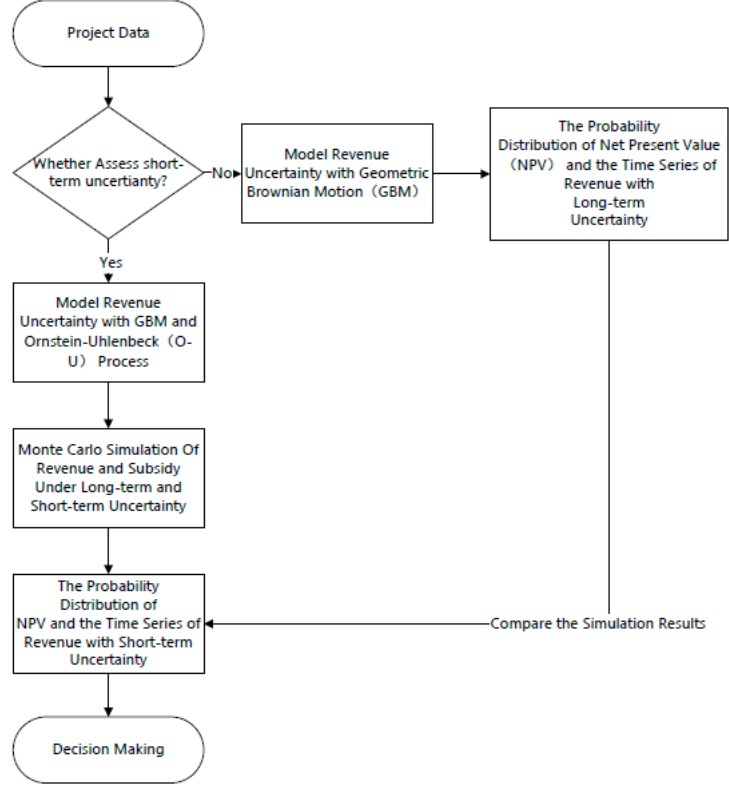

**Figure 2.** Flowchart of the proposed mixed model.

## 3. Case Study

### 3.1. Illustrative Case Example

This section presents the noise-based option model for the case of the PPP sewage treatment project in China [41]. Hypothetical cases are commonly used for demonstration in real option literature [31,42]. In this article, we introduced some additional inputs, such as the volatility of short-term uncertainty, mean-reverting rate and mean value of noise. The values of the parameters used in the evaluation are shown in Table 1.

**Table 1.** Overview of the PPP Sewage Treatment case study main input parameters.

| Project Parameters | Symbol | Value |
|---|---|---|
| Construction cost | I | 101.18 million |
| Operation and maintain costs | $C_t$ | 2 million |
| Construction period | $T_c$ | 2 years |
| Operation period | $T_o$ | 20 years |
| Toll Rate | P | 0.8285 yuan |
| Design capacity | Q | 80,000 T |
| Discount rate | r | 6.90% |
| Upper threshold | $\theta_{max}$ | 110% |
| Lower threshold | $\theta_{min}$ | 70% |
| Annual growth rate | $\mu_X$ | 8.40% |
| Long-term volatility | $\sigma_X$ | 6.20% |
| Mean reverting rate | k | 0.1 |
| Short-term volatility | $\sigma_Y$ | 0.2 |
| Mean reversion value | u | −1 |

Note: Values in millions CNY (Chinese yuan).

The construction period and operation period are 1 and 20 years respectively. The basic data-like construction cost is determined by the design, and the toll rate is determined by the local government. The expected annual growth rate of the sewage treatment value over the life of the project is 8.4%, and the long-term volatility of the annual growth rate of the sewage treatment value is 6.2%. The data-like operation and maintenance costs, annual growth rate and long-term volatility are calculated by the historical monthly cash flow data of a similar project. The discount rate is 6.9%, estimated from the relevant data of the 10-year bond yield to maturity of the local government. The mean reverting rate, short-term volatility and mean reversion value are supposed to be estimated by the cash flow data of a similar project; however, here we use the given value to make the sensitivity analysis. The mean reverting rate is 0.1, short-term volatility is 0.2 and the mean reversion value is −1.

### 3.2. Results and Discussion

We now use a Monte Carlo simulation calculated by 10,000 trials to compare the probability distributions of the subsidy without noise and with O-U process noise to study the effect of the short-term uncertainty on the subsidy.

The simulated revenue is illustrated in Figure 3: the long right-left tail shows that the actual revenue is lower than its forecasts. This can be one evidence that shows that short term uncertainty affects the standard deviation, skewness, and kurtosis of revenue [43]. In other words, short-term uncertainties determine the location and the shape of the distribution.

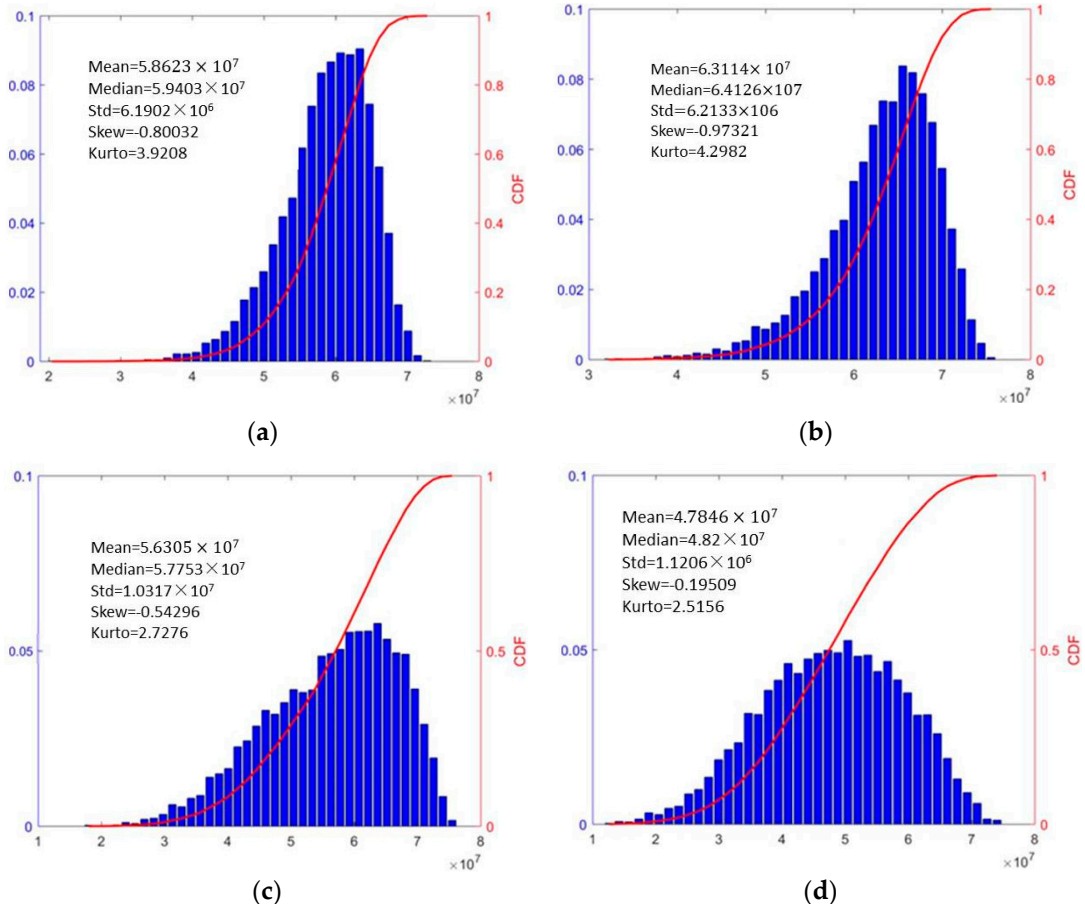

**Figure 3.** The distribution of the private revenue without and with noise. (The *x*-axis shows the NPV of the private sector. The *y*-axis shows the frequency. Subplot **a**: Revenue without short-term uncertainty. Subplot **b**, **c**, **d** explicitly show the revenue with short-term uncertainty, and the mean reversion values are 1, 0, and −1 respectively).

These results highlight the significant differences in the option value without and with short-term uncertainty. The mean value of the corresponding Net Present Value (NPV) was $5.86 \times 10^7$ CNY without the short-term uncertainty. The NPV increased to $6.31 \times 10^7$ CNY when the mean value of the short-term uncertainty is positive (u = 1). Additionally, as the mean value of the short-term uncertainty changed to −1, the NPV decreased from $5.86 \times 10^7$ CNY to $4.78 \times 10^7$ CNY. From the above statistic, we can see that the revenue increases by about 7.6% when the mean value of the short-term uncertainty increased to 1, compared with the revenue without the short-term uncertainty. Furthermore, the revenue decreases by about 18.4% when the mean value of the short-term uncertainty decreased to −1, compared with the revenue without the short-term uncertainty. The results show that short-term uncertainty can have both positive and negative effects on the revenue. The historical project experiencing negative noise shows the reduction in the expected revenue of the new project, while the positive noise depicts an increment in the expected revenue in a new project. This is consistent with the study by Hong et al. [44], which states that the effects of short-term uncertainties on the real option value of the project could be positive or negative. Empirical evidence demonstrated that the demand deviation was about 20% in public works projects [45]. The deviations in revenue estimated in this paper are consistent with this empirical evidence.

The simulated government subsidy is illustrated in Figure 4. The mean value of the government subsidy was $3.48 \times 10^7$ CNY without short-term uncertainty. The subsidy increased to $7.51 \times 10^7$ CNY when the mean value of the short-term uncertainty is positive (u = 1). Additionally, as the mean value of the short-term uncertainty changed from 0 to −1, the subsidy decreased from $3.49 \times 10^7$ CNY to $-2.99 \times 10^7$ CNY. An increase in subsidy could lead to high-performance costs for the government and prevents the PPP contract from being successfully completed [46]. Over-valuation caused by not taking into account short-term uncertainty may explain unprofitable projects [27]. From the above statistic, we can see that the government subsidy increased about 115% when the mean value of short-term uncertainty increased to 1, compared with the revenue without the short-term uncertainty. Additionally, the government subsidy decreased about 185% when the mean value of the short-term uncertainty decreased to −1, compared with the revenue without the short-term uncertainty.

Compared with the change in revenue, the government subsidy is more sensitive to the change of short-term uncertainty. This is because, owing to the protection of the collar option mechanism, the private revenue will stay between regions. Shan pointed out that the collar option mechanism reduces downside losses and guarantees basic returns to stakeholders [32]. However, the value differences caused by the short-term uncertainty will be undertaken by the government.

Figure 5 shows the cumulative distribution of the government subsidy. The change of the short-term uncertainty has influences on the government subsidy. Increasing the mean value of the short-term uncertainty results in a clear leftward shift relative to the long-term uncertainty. This is because the rise of the mean value leads to an increase of the short-term uncertainty revenue, which will increase the actual revenue and decrease the government subsidy accordingly. The results are consistent with the actual condition. The reserve revenue was generated using the deductive approach, using increments generated through Equations (5) and (8), respectively.

The sensitivity analysis is performed to explain the impact of different features of noise on the real option value. A one-factor-at-a-time (OFAT) approach is applied to study how the short-term uncertainty parameters (e.g., mean reversion value and reverting speed.) influence the estimated revenue. Figure 6 shows one sample of all 10,000 paths that were simulated. Subplot a is the base sample path and Subplot b, c and d in Figure 6 use different values for the short-term uncertainty parameters, which are listed in Table 2. In the sensitivity analysis below, the short-term uncertainty parameters used are assumed. The value of these short-term uncertainty parameters should be calculated from the actual sewage treatment value.

**Table 2.** Values for the short-term uncertainty parameters in the sensitivity analysis.

| Parameters | Subplot a | Subplot b | Subplot c | Subplot d |
|---|---|---|---|---|
| Reverting speed (k) | 0.1 | 1 | 0.1 | 0.1 |
| Mean value (u) | −1 | −1 | 1 | −1 |
| Volatility ($\sigma_y$) | 0.5 | 0.5 | 0.5 | 1 |

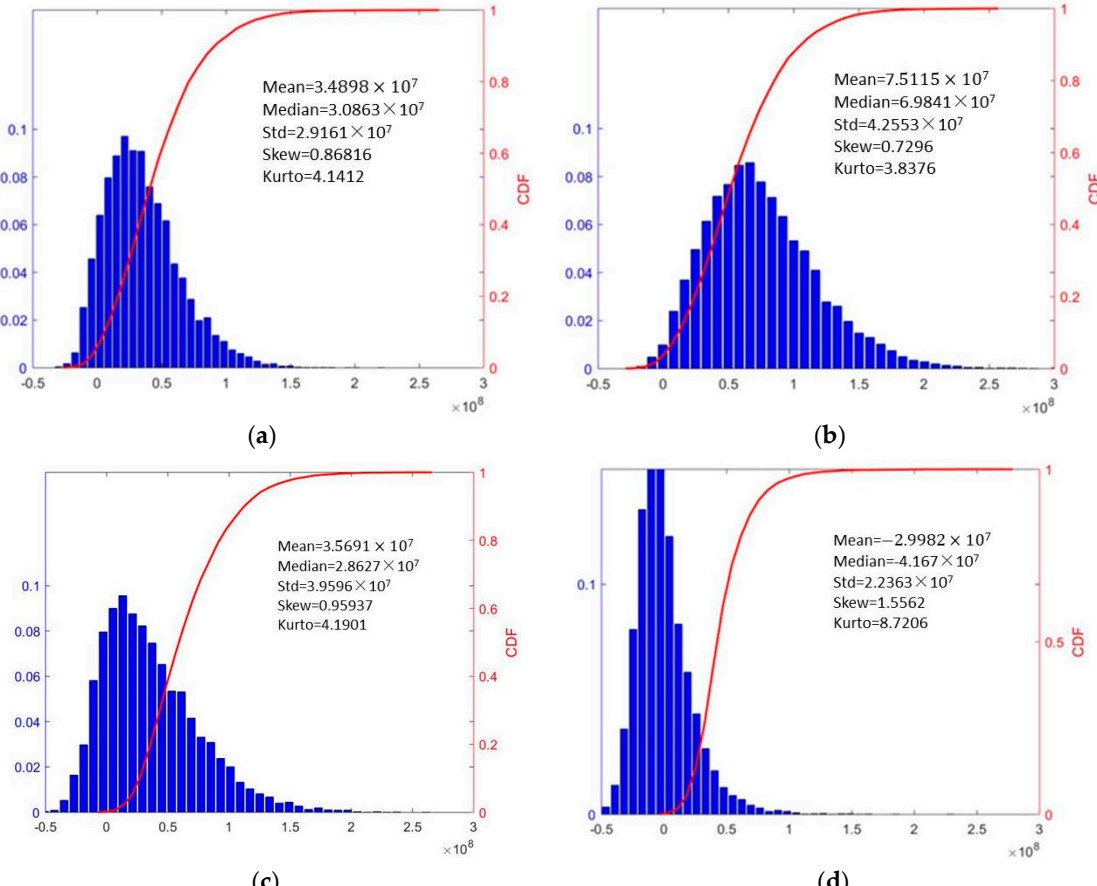

**Figure 4.** The distribution of the government subsidy without and with noise. (The *x*-axis shows the government subsidy. The *y*-axis shows the frequency. Subplot **a**: Subsidy without short-term uncertainty. Subplot **b**, **c**, **d** explicitly show the subsidy with short-term uncertainty, and the mean reversion values are 1, 0, and −1 respectively).

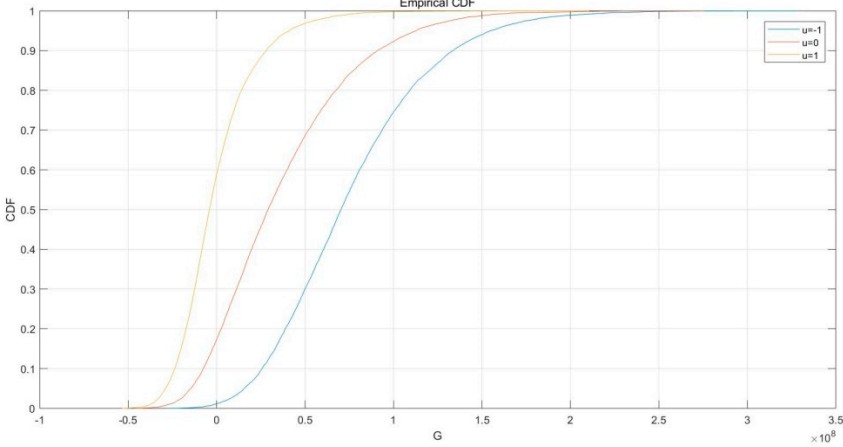

**Figure 5.** Cumulative Distributions Function (CDF) of the revenue. (u is the mean value).

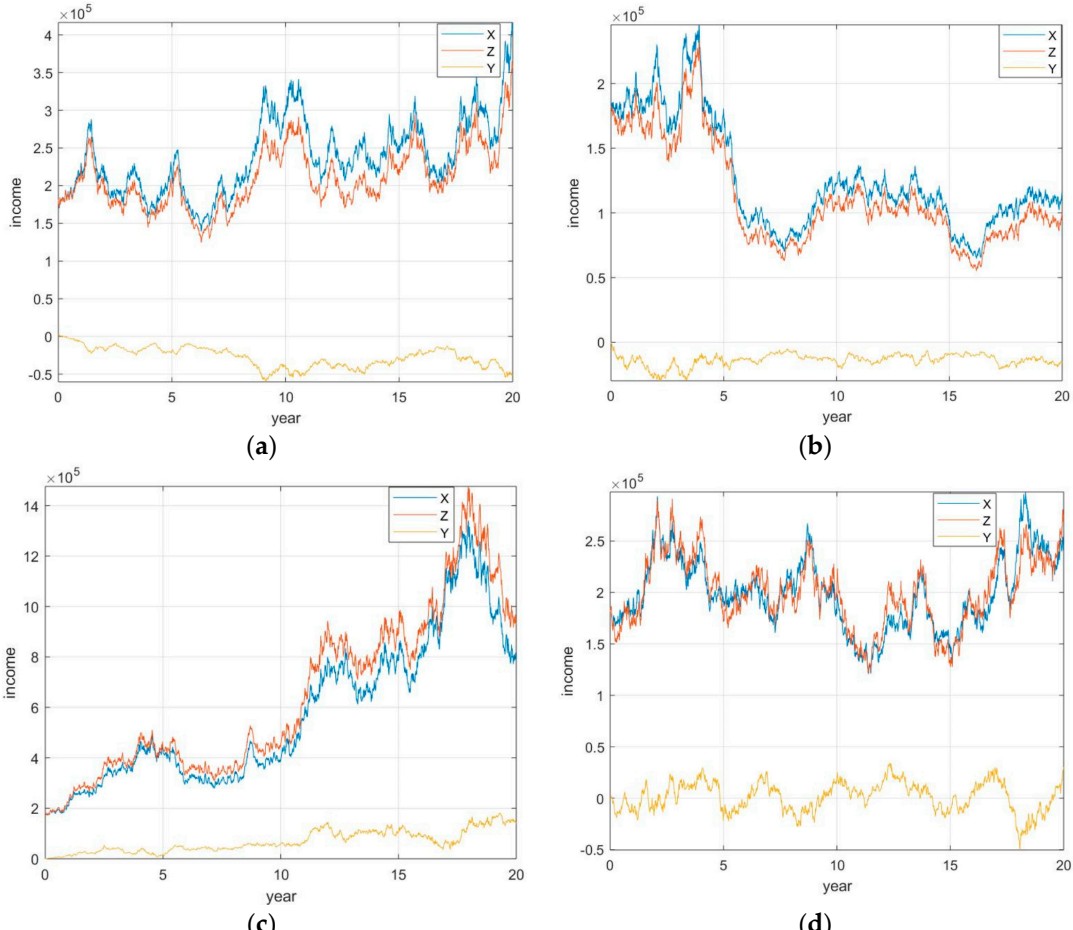

**Figure 6.** Time series simulation of revenues (The *x*-axis shows the timeline of the operation period. The *y*-axis shows the revenue. Subplot **a**: Base sample. Subplot **b**: High rate of the mean reversion, k. Subplot **c**: High rate of the mean value u. Subplot **d**: High rate of the noise volatility term, $\sigma_y$. This figure indicates that the time series with an alternative parameter are different from each other. The blue lines X represent the estimated revenue, the red lines Z represent the actual revenue, while the yellow lines Y represent the basis revenue. Subplots **a**, **b**, **c** and **d** represent the time trends of three scenarios. Note the different scales in the *y*-axis).

Figure 6 plots the typical paths of the 25-year time series of the estimated revenue, actual revenue and noise revenue with alternative parameter values. The processes used to produce these paths are defined in Equations (5), (6), and (8).

The subplot is a base sample. It shows that the noise is persistently negative from years 1 to 20, resulting in actual revenues that are below the estimation value. From years 15 to 20, the noise revenue becomes negative, resulting in actual revenues that are below the estimated revenue. Because the noise stays near zero between the years 10–15, the actual revenue tracks the estimation value relatively closely during that period. The mean reversion in the noise process keeps the actual revenue from wandering too far away from the estimated revenue.

We can see that the noise revenue fluctuates around a mean value which is near zero and that the actual revenue has a very significant correlation with the estimated revenue. When Y ≥ 0, the actual revenue Z is parallel to the main direction of the estimated revenue X, which means that when the external conditions produce a positive effect on the project, the actual revenue exceeds the estimated revenue. When Y ≤ 0, the movement of the actual revenue Z and the estimated revenue are in opposite directions, which means that when the external conditions produce a negative effect on the project, the actual revenue is smaller than the estimated revenue.

Subplots b, c, and d show revenue changes by using alternative parameters. Subplot b increases the reverting speed. The value of k determines the speed at which the noise dissipates. k = 0 means that the noise follows a random walk and that there is no mean reversion. As the value of k increases, the value of Y goes back near to the mean value. We can see from Subplot b that the value of Y is stable and the approach to zero indicates that the market is stable. The reverting speed and noise values showed the reverse changes in the relationship. As a result, the actual revenue tracks the estimated revenue much more closely than in Subplot a.

As an indicator of the intensity for government control policy, the reverting speed k has a positive effect on the revenue, indicating that when the market is volatile, it is better for the government to make a quick and aggressive response to the incident, such as promulgating adjusting the policy to achieve the aim of inducing the noise revenue and stabilizing the government subsidy. Otherwise, the private sector should deliberate over the government administrative level before making a decision. Therefore, the government should establish a special institution and arrange professional staff responsible for the execution of the PPP project, and there are thorough processes and coping mechanisms to guide the participants. Kou and Luo's research noted that the mean-reverting speed has an impact on asset pricing [47]. The simulated result of Zhao et al. [48] showed that the call and put option values will decrease as the speed of reversion increases.

Subplot c increases the mean reversion value u Compared with Subplot a, Subplot b shows that when u = 1, the short-term uncertainty is basically larger than zero after year 5. u is determined by the differences of the actual revenue and estimated revenue of the historical data. As a market indicator, u ≥ 0 means that the noise has a positive effect on the revenue, indicating that the market is profitable for the private sector for a long time in the past, and the private sector prefers to invest now, and vice versa. It is important for the government to collect and supervise the revenue data of the PPP projects which are operating. Historical data can be used to conduct a performance assessment of projects and analyze the basic tendency of the PPP project planned by the government and private sector. This viewpoint is also supported by Roehrich et al. [49], according to which transparent historical data is important for risks ex-ante quantification.

The trends observed in the results are also in agreement with the literature that estimates the impact of uncertainties on the revenue of revenue systems with similar characteristics as PPP projects. The study of Bekaert et al. [50] is one such literature that estimated the effect of volatility on revenue.

The expected revenue can be realized in most developed countries because the majority of them have sound PPP policies; however, this is not the case in most developing countries [51].

Subplot d increases the noise volatility term $\sigma_y$. The results show that there is an over-evaluation in the years 2–7, while there is an undervaluation in the years 13–15. The noise volatility indicates the occurrence frequency of the market risk and the perfection level of the market regulation and policies, such as the extent of information transparency and disclosure. When the market risk increases, the actual revenue will obviously fluctuate; it is not good for the government to smooth its annual subsidy. This is because the negative mean value of noise has a negative influence on the asset value, and short-term uncertainty is going to revert based on the policy adjustment. The noise volatility becomes higher as the market becomes more un-transparent. Generally, in a standard real options model, a higher volatility implies a higher option value of the real asset; however, in the case of noise there is a different appearance: a higher volatility leads to severe over- and undervaluation. The simulation results of Wang et al. [52] proved that an increase in the volatility of noise will lead to an enhanced unpredictability of the real option value and risks.

Therefore, a normative transaction discipline is important, and information disclosures reduce the noise value.

In summary, short-term uncertainty parameters such as the reverting speed and mean value have a positive effect on the revenue, while noise volatility leads to the over- and under-evaluation of the revenue. Among all of the parameters, the mean value determines the general direction of the noise movement. The government and private sector should pay attention to short-term uncertainties

because the introduction of a short-term uncertainty will give more accurate results of the projected revenue. This may lead to different investment decisions [28,53].

This article has the following limitations. (1) The model needs to be applied to real PPP projects to validate the applicability of the model in real projects; (2) The assumptions should be taken into account when interpreting the results of the article; and (3) Behavioral factors also have impacts on the success or failure of PPP projects. Hence, these factors should be assessed through further research.

The quantitative relationship between uncertainty factors and the noise level will be investigated in future studies. This will help the decision-makers to make a more precise decision.

## 4. Conclusions

This article's general objective is motivated by the fact that there are significant differences between the theoretical predictions of PPP projects' revenues and their real value.

This article identifies the effect of short-term uncertainty in the application of ROT. We focus on the study of the PPP sewage treatment revenue, which follows a mixed process of long-term uncertainty together with short-term uncertainty. The presence of short-term uncertainty complicates the valuation of the PPP sewage treatment revenue and decision making.

The simulated results show that a mean-reverting short-term uncertainty may significantly change the mean value and the distribution of revenue and government subsidy. The government subsidy is more sensitive to the change of short-term uncertainty.

The annual revenue may significantly fluctuate because of unpredicted events during the operation period. The discrepancy between the estimated revenue and the actual revenue will lead to an unprepared government subsidy, and the total subsidies may significantly deviate from the government infrastructure investment budget with the effect of the time value of money.

The government needs to take adjustment actions to maintain the stability and gradually depreciate the disturbance by short-term uncertainty to the actual revenue when there is an obvious deviation in the PPP market.

It is necessary to establish an information feedback mechanism to evaluate the accuracy of the estimation in the planning stage, and this feedback mechanism may help to make precise decisions to PPP projects by considering the level of noise caused by uncertainty and propose feasible measures to improve the subsequent operation performance.

Generally, the simulated results of this paper contribute insights that can be used to improve the BS model. The traditional B-S model might incorporate errors because of its failure to incorporate short term uncertainty. This article combines the random walk and mean reverting features of the revenue. This is a more practical approach and improves the accuracy of revenue forecasting.

**Author Contributions:** Q.L. and Q.G. proposed the research ideas and methods of the manuscript. Q.L. wrote the original draft. S.W. wrote the code. Z.L and D.M.D. put forward the review and editing to the paper. F.J. is responsible for creating figures and forms.

**Funding:** This research was partially funded by Natural Sciences and Engineering Research Council of Canada (NSERC.DG NO.443130).

**Conflicts of Interest:** The authors declare no conflict of interest.

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
