# Peer review of "Effects of Short-Term Uncertainties on the Revenue Estimation of PPP Sewage Treatment Projects"

_water, doi:10.3390/w11061203_

Round 1
Reviewer 1 Report
1. Literature should be quoted in accordance with the guidelines of the journal.
2. There is no discussion of results in chapter 3.2. In this chapter are only results. In this chapter, it is necessary to rely on results similar to own research and compare them with results from other publications.
E.g. Water 2019, 11, 873; doi: 10.3390 / w11050873 "Application of the Mathematical Simulation Methods for the Assessment of the Wastewater Treatment Plant"
Author Response
Point 1: Literature should be quoted in accordance with the guidelines of the journal.
Response 1: We have quoted the literatures in accordance with the guidelines of the journal.
Point 1: There is no discussion of results in chapter 3.2. In this chapter are only results. In this chapter, it is necessary to rely on results similar to own research and compare them with results from other publications.
E.g. Water 2019, 11, 873; doi: 10.3390 / w11050873 "Application of the Mathematical Simulation Methods for the Assessment of the Wastewater Treatment Plant"
Response2: The discussion section of the paper is modified and the added part is highlighted in the manuscript.
Reviewer 2 Report
The paper is about a very interesting and important topic. The risk factor(s) within PPP are relevant factors for the success of PPP. Albeit it is hard to predict many of these risks. Risks arise in every phase of PPP.
1. In line 45 ff. you state, that PPP allow the government to mitigate their financial burden. You realize that the government has to pay but not immediately. Therefore a PPP is often “only” a credit from private to public sector. Often later generations have more financial burden due to the PPP. This should be kept in mind. You state that the public sector has to attract the private sector for these “unprofitable infrastructure projects”. In many cases they are not that unprofitable for the private sector but they make it unprofitable for the public sector. The question is about the value for all of the partners.
2. As risk is such an important factor for the success of PPP, it would be helpful for the reader (maybe not all readers are familiar with PPP) to get more information about the different risks and their occurance. Especially European literature gives in depth insights into risks within PPP. You can differentiate risks in the several phases of a PPP. Therefore you should describe these issues a bit more detailed. In line 56 ff. you mention different risks but which of these risks are important and are considered within your model. How can you evaluate these risks. There exist estimations for some of them. The risks are specific for different models of PPP. Which models do you consider/include in your paper?
3. What is with behavioral aspects within PPP. Many PPP projects do not work, as the private partner underestimate some costs prior to the project to get the tender and afterwards costs are adjusted to a higher level. Are different aims of public and private sector considered? Often many risks remain with the public sector, as the private sector is not willing to incorporate them. In this case the project is not always valuable for the public sector. If you talk about value of a PPP. Value for whom? For the private sector? For the public sector? For citizens?
4. There is no real information about the PPP model. This would be important to relate the risks and the assignment of responsibilities. There exist many PPP models and each is different from risk allocation. This has to be taken into consideration.
5. In line 49 f. you mention the infrastructure construction. What about the maintenance? Many PPP projects combine construction and maintenance of the infrastructure. Is that also included? (you only mention that in table 1 with the costs and operation period). But there exist also models with split maintenance. What about these?
6. There is no real discussion of the findings. The findings are described, but there should be a more detailed discussion of these findings against existing literature. Which factors can be influenced etc.
7. In your limitations you state, that this model now has to be applied to a real project. Can you describe, how you can do this? Do you also want to use this model for other projects than sewage?
8. Overall, it is a good idea, to use this model for PPP with respect to risks and uncertainties, but the paper could be improved:
- Description of phases, risks, uncertainties in different PPP models (which model is used for sewage?)
- There is not so much international literature of PPP in the paper. Especially in the European context (as some of the European countries are pioneers in PPP/PFI), this literature should be considered.
Author Response
Response to Reviewer 2 Comments
Point 1: In line 45 ff. you state, that PPP allow the government to mitigate their financial burden. You realize that the government has to pay but not immediately. Therefore a PPP is often “only” a credit from private to public sector. Often later generations have more financial burden due to the PPP. This should be kept in mind. You state that the public sector has to attract the private sector for these “unprofitable infrastructure projects”. In many cases they are not that unprofitable for the private sector but they make it unprofitable for the public sector. The question is about the value for all of the partners.
Response 1: The purpose of PPP schemes is to alleviate the government financial burden, we agree that due to the complexity of the PPP schemes and the uncertainties in the future there is a possibility that they could end up being a financial burden for the government. We mentioned these are pitfalls of the PPP schemes in the introduction section of the revised manuscript, see Line 55-61.
Point 2: As risk is such an important factor for the success of PPP, it would be helpful for the reader (maybe not all readers are familiar with PPP) to get more information about the different risks and their occurrence. Especially European literature gives in depth insights into risks within PPP. You can differentiate risks in the several phases of a PPP. Therefore you should describe these issues a bit more detailed. In line 56 ff. you mention different risks but which of these risks are important and are considered within your model. How can you evaluate these risks. There exist estimations for some of them. The risks are specific for different models of PPP. Which models do you consider/include in your paper?
Response2: According to the reviewer advice we have included more details about the details about the classification of uncertainties in the PPP projects in Introduction, see Line66-74.
Point 3: What is with behavioral aspects within PPP. Many PPP projects do not work, as the private partner underestimate some costs prior to the project to get the tender and afterwards costs are adjusted to a higher level. Are different aims of public and private sector considered?
Often many risks remain with the public sector, as the private sector is not willing to incorporate them. In this case the project is not always valuable for the public sector. If you talk about value of a PPP. Value for whom? For the private sector? For the public sector? For citizens?
Response 3: We acknowledge that behavioral factors also have impacts on the success or failure of PPP projects, but they are not considered in this paper and are beyond the scope of this paper. This is listed as one of the limitations of the paper that need to be addressed through further research. PPP projects are aimed to solve current infrastructural problems communities face. If planned very well, for example like the projects in the developed countries their future financial burden on the government can be mitigated. Hence with good planning these initiatives can be very successful and beneficial to all stake holders creating a win-win scenario.
Point 4: There is no real information about the PPP model. This would be important to relate the risks and the assignment of responsibilities. There exist many PPP models and each is different from risk allocation. This has to be taken into consideration.
Response 4: In this research, we don’t study the risk allocation. We agree the different ways PPP projects are arranged mainly depend on the risk allocation. We mainly focused on how to improve the accuracy of predicting the future revenue. We will take this into consideration and address it through future research. We thank the reviewer for the suggestion.
Point 5: In line 49. you mention the infrastructure construction. What about the maintenance? Many PPP projects combine construction and maintenance of the infrastructure. Is that also included? (you only mention that in table 1 with the costs and operation period). But there exist also models with split maintenance. What about these?
Response: 5 we thank the reviewer for the Point . we agree that the maintenance costs also affected by various short term uncertainties, but in this paper we did not take into account the maintain ace cost because we our main goal was to show how the noise affects the predicted revenue value and this can also be extend to include the maintenance costs. We will address it through further research.
Point 6: There is no real discussion of the findings. The findings are described, but there should be a more detailed discussion of these findings against existing literature. Which factors can be influenced etc.
Response 6: According to the reviewer advice we have modified the results and discussion section of the paper by supporting the results with relevant literature, see Line 245-251,Line 264-266,Line 273-274,Line 332-334,Line 353-359,Line 369-371,Line 377-380.
Point 7: In your limitations you state, that this model now has to be applied to a real project. Can you describe how you can do this? Do you also want to use this model for other projects than sewage?
Response 7: The application of this model to real projects will verify the accuracy of the model. This could also be as source of insights to further modify the model.
Point 8: Overall, it is a good idea, to use this model for PPP with respect to risks and uncertainties, but the paper could be improved:
- Description of phases, risks, uncertainties in different PPP models (which model is used for sewage?)
Response 8: This study didn’t focus on the different ways to form PPP schemes but rather on simulation of revenue uncertainty. And the uncertainties are classified by the time duration of the influence on project revenue/ government subsidy. Detailed descriptions are given in the introduction according to the reviewer advice.
Point 9: There is not so much international literature of PPP in the paper. Especially in the European context (as some of the European countries are pioneers in PPP/PFI), this literature should be considered.
Response: 9 International literatures are included in the paper according to the reviewer’s Point s. See Line 55-61, Line 66-74.
Reviewer 3 Report
The paper addresses an interesting and very important research area, investigating how uncertainty in revenue estimation impacts public-private partnership projects. The authors draw on some prior studies, but a much more critical literature analysis is needed to strengthen the paper’s argument and draw out the (theory) gaps they seek to address. Also, the paper needs to be present much stronger discussion and conclusion sections in order to offer value to the reader. Overall, the manuscript makes some very interesting points and I realize that a lot of work went into this study. Nevertheless, I see room for improvement which will help to enhance clarity, readability, practical and theoretical contributions. The following paragraphs address each section of the paper in more detail and provide suggestions on how to revise the paper.
Major concerns:
Introduction:
While the author(s) establish some links to some extant literature, author(s) need to establish a more coherent framework for the overall paper. That means, the introduction should clearly indicate the need for this paper in relation to extant research studies. The authors need to draw out further the gap(s) they seek to address with regards to extant studies. Here, the authors need to more clearly link PPPs and key extant studies (e.g. James Barlow, Nigel Caldwell, Ilze Kivleniece). The authors may also consider linking their work to recent discussions regarding sustainable development and concepts of sustainability (e.g. Stefan Hoejmose, Helen Walker; please see some suggestions for further studies in the reference list at the end of my comments).
Conceptual background & Theoretical development:
The authors need to establish this section (or present a much longer introduction section which clearly links their research to extant studies) some clearer links to extant PPP literature (e.g. please see suggested key references) to guide the reader. This would help to clearly establish the basis of this paper. The author(s) should clearly draw out the benefits and limitations of PPPs and issues around the management of such long-term relationships to drive social responsibility and economic impact. This should then be linked to discussions around value for money, risks/uncertainties and sustainable development as key drivers. A much clearer positioning of the current study will help to further guide the reader and draw out gap(s) in extant studies.
Discussions and Conclusions:
Derived from a conceptual background section which did not clearly draw out the gaps the paper seeks to address, the discussion (missing) and conclusion sections do offer no additional value to the reader as it stands. The authors need to offer more fine-grained results here and discuss what they intended to find out in the introduction section (link to research question; overall aim of the paper). Overall, the authors need to clearly draw out what the theoretical contributions are and how they add to the existing body of knowledge. This section also needs to clear link back to extant studies to offer some clear value to the reader.
Useful references:
Brammer, S. and Walker, H. L. 2011. Sustainable procurement in the public sector: an international comparative study. International Journal of Operations & Production Management 31(4), pp. 452-476.
Roehrich et al. (2014). Are public-private partnerships a healthy option? A systematic literature review. Social Science & Medicine, Vol. 113, pp. 110-119.
Touboulic, A. and Walker, H. L. 2015. Theories in sustainable supply chain management: a structured literature review. International Journal of Physical Distribution & Logistics Management 45(1/2), pp. 16-42.
Zheng, J.; Roehrich, J.K. and Lewis, M.A. (2008). The dynamics of contractual and relational governance: Evidence from long-term public-private procurement arrangements. Journal of Purchasing and Supply Management, Vol. 14 No. 1, pp. 43-54.
Author Response
Response to Reviewer 3 Comments
Reviewer 3
Point 1: Introduction:
While the author(s) establish some links to some extant literature, author(s) need to establish a more coherent framework for the overall paper. That means, the introduction should clearly indicate the need for this paper in relation to studies. The authors need to draw out further the gap(s) they seek to address with regards to extant studies. Here, the authors need to more clearly link PPPs and key extant studies (e.g. James Barlow, Nigel Caldwell, Ilze Kivleniece). The authors may also consider linking their work to recent discussions regarding sustainable development and concepts of sustainability (e.g. Stefan Hoejmose, Helen Walker; please see some suggestions for further studies in the reference list at the end of my Point s).
Response1:We have established the framework for the overall paper according to the reviewer’s suggestion, see Figure 2. We mentioned very briefly how the study is linked to sustainable development in Line 97 and recommended literature is was cited.
Point 2: Conceptual background & Theoretical development:
The authors need to establish this section (or present a much longer introduction section which clearly links their research to extant studies) some clearer links to extant PPP literature (e.g. please see suggested key references) to guide the reader. This would help to clearly establish the basis of this paper. The author(s) should clearly draw out the benefits and limitations of PPPs and issues around the management of such long-term relationships to drive social responsibility and economic impact. This should then be linked to discussions around value for money, risks/uncertainties and sustainable development as key drivers. A much clearer positioning of the current study will help to further guide the reader and draw out gap(s) in extant studies.
Response2: We have modified the introduction section of the paper according the suggestion. The pitfalls of PPP scheme and the influence of short-term uncertainty on the economic impact and performance of the projects are stated in detail, see line 95-97.
Point 3: Discussions and Conclusions:
Derived from a conceptual background section which did not clearly draw out the gaps the paper seeks to address, the discussion (missing) and conclusion sections do offer no additional value to the reader as it stands. The authors need to offer more fine-grained results here and discuss what they intended to find out in the introduction section (link to research question; overall aim of the paper). Overall, the authors need to clearly draw out what the theoretical contributions are and how they add to the existing body of knowledge. This section also needs to clear link back to extant studies to offer some clear value to the reader.
Useful references:
Brammer, S. and Walker, H. L. 2011. Sustainable procurement in the public sector: an international comparative study. International Journal of Operations & Production Management 31(4), pp. 452-476.
Roehrich et al. (2014). Are public-private partnerships a healthy option? A systematic literature review. Social Science & Medicine, Vol. 113, pp. 110-119.
Touboulic, A. and Walker, H. L. 2015. Theories in sustainable supply chain management: a structured literature review. International Journal of Physical Distribution & Logistics Management 45(1/2), pp. 16-42.
Zheng, J.; Roehrich, J.K. and Lewis, M.A. (2008). The dynamics of contractual and relational governance: Evidence from long-term public-private procurement arrangements. Journal of Purchasing and Supply Management, Vol. 14 No. 1, pp. 43-54.
Response 3: The main objective of the paper is to study the influence of noise in predicted revenue of PPP projects. The introduction section of the paper is modified to show the research gaps and the main contribution of the pepper according to the reviewer’s Point. A limitation from the previous studies is that, although they gave their identified uncertainty factors and demonstrated the existence of short-term uncertainty, they fail to evaluate long-term and short-term uncertainties in one statistical model and clearly explain the reasons why the theoretical value of PPP subsidy deviate from the real value. Is this deviation caused by short-term uncertainty? If it is, how does short-term uncertainty influence the real option value? The current study aims to fill this research gap.in addition the relevant references recommended by the reviewer are included in the article.
Round 2
Reviewer 1 Report
In this version, the article can be published
Author Response
We thank the reviewer for the comments and suggestions.
Reviewer 2 Report
Thank you for revising the paper. Some minor advices:
Line 60: I would not call PPP a scheme.
Line 80: I think it can be discussed, if financial crisis and war are short-term uncertainties. They also lead to a long-term trend of the volatility.
Line 161, 174, 176, 179: There stands “Error, Reference Source not found.” This should be corrected
To the data used in the case study: it should be explained how you estimate these values. Are these experiences from other cases or deduced from literature? You state that this is a hypothetical case, but for the reader it should be comprehensible were the data comes from. A short description of how these values are developed would be helpful.
Line 233/234: “The long right-left tail represents actual revenue that less than its forecasts.” Is there a word missing?
Author Response
Reviewer 2
Comment 1: Line 60: I would not call PPP a scheme.
Response:The word scheme is replaced by arrangements according to the reviewer’s suggestion.
Comment 2: Line 80: I think it can be discussed, if financial crisis and war are short-term uncertainties. They also lead to a long-term trend of the volatility.
Response: The authors agree with the reviewer’s comments and replaced the terms “financial crisis” and “war” with “short term financial crisis and political instability”.
Comment 3: Line 161, 174, 176, 179: There stands “Error, Reference Source not found.” This should be corrected
Response: Modified according to the reviewer’s comments
Comment 4: To the data used in the case study: it should be explained how you estimate these values. Are these experiences from other cases or deduced from literature? You state that this is a hypothetical case, but for the reader it should be comprehensible were the data comes from. A short description of how these values are developed would be helpful.
Response: This case is cited from Ning Liu’s research and it was real case in China. The basic data like construction cost is determined by the design and toll rate is determined by the local government. Data like operation and maintain costs, annual growth rate and long-term volatility are calculated by the historical monthly cash flow data of similar project. Discount rate is estimated from the relevant data of the 10-year bond yield to maturity of local government. The mean reverting rate, short-term volatility and mean reversion value are supposed to be estimated by cash flow data of similar project, however , here we use given value to make the sensitivity analysis.
Comment 5: Line 233/234: “The long right-left tail represents actual revenue that less than its forecasts.” Is there a word missing?
Response: We have corrected this sentence as “The simulated revenue is illustrated in Figures 3, the long right-left tail shows that the actual revenue is less than its forecasts”.
Reviewer 3 Report
The paper addresses an interesting and very important research area, investigating how uncertainty in revenue estimation impacts public-private partnership projects. The authors have only superficially addressed my comments from round 1 and thus the current study (study’s argument) has not been improved. Please find below core comments from the previous round and additional comments for this round. Please ensure to clearly address these comments as this will help to further improve your study’s argument and positioning in the wider PPP literature. The following paragraphs address each section of the paper in more detail and provide suggestions on how to revise the paper.
Major concerns:
Introduction:
While the author(s) establish some links to some extant literature, author(s) need to establish a more coherent framework for the overall paper. That means, the introduction should clearly indicate the need for this paper in relation to extant research studies. The authors need to draw out further the gap(s) they seek to address with regards to extant studies. Here, the authors need to more clearly link PPPs and key extant studies (e.g. James Barlow, Nigel Caldwell, Ilze Kivleniece). The authors may also consider linking their work to recent discussions regarding sustainable development and concepts of sustainability (e.g. Stefan Hoejmose, Helen Walker; please see some suggestions for further studies in the reference list at the end of my comments).
Conceptual background & Theoretical development:
This section needs to be established to clearly position the authors’ current study. There needs to be a clear link to extant studies (please see reference list for some useful suggestions) to ground the current study. The authors need to establish this section (or present a much longer introduction section which clearly links their research to extant studies) some clearer links to extant PPP literature (e.g. please see suggested key references) to guide the reader. This would help to clearly establish the basis of this paper. The author(s) should clearly draw out the benefits and limitations of PPPs and issues around the management of such long-term relationships to drive social responsibility and economic impact. This should then be linked to discussions around value for money, risks/uncertainties and sustainable development as key drivers. A much clearer positioning of the current study will help to further guide the reader and draw out gap(s) in extant studies.
Discussions and Conclusions:
This still needs substantial improvements. Derived from a conceptual background section which did not clearly draw out the gaps the paper seeks to address, the discussion (missing) and conclusion sections do offer no additional value to the reader as it stands. The authors need to offer more fine-grained results here and discuss what they intended to find out in the introduction section (link to research question; overall aim of the paper). Overall, the authors need to clearly draw out what the theoretical contributions are and how they add to the existing body of knowledge. This section also needs to clear link back to extant studies to offer some clear value to the reader.
Useful references:
Brammer, S. and Walker, H. L. 2011. Sustainable procurement in the public sector: an international comparative study. International Journal of Operations & Production Management 31(4), pp. 452-476.
Roehrich et al. (2014). Are public-private partnerships a healthy option? A systematic literature review. Social Science & Medicine, Vol. 113, pp. 110-119.
Touboulic, A. and Walker, H. L. 2015. Theories in sustainable supply chain management: a structured literature review. International Journal of Physical Distribution & Logistics Management 45(1/2), pp. 16-42.
Zheng, J.; Roehrich, J.K. and Lewis, M.A. (2008). The dynamics of contractual and relational governance: Evidence from long-term public-private procurement arrangements. Journal of Purchasing and Supply Management, Vol. 14 No. 1, pp. 43-54.
Author Response
Reviewer 3
Major concerns:
Comment: 1 Introduction:
While the author(s) establish some links to some extant literature, author(s) need to establish a more coherent framework for the overall paper. That means, the introduction should clearly indicate the need for this paper in relation to extant research studies. The authors need to draw out further the gap(s) they seek to address with regards to extant studies. Here, the authors need to more clearly link PPPs and key extant studies (e.g. James Barlow, Nigel Caldwell, Ilze Kivleniece). The authors may also consider linking their work to recent discussions regarding sustainable development and concepts of sustainability (e.g. Stefan Hoejmose, Helen Walker; please see some suggestions for further studies in the reference list at the end of my comments).
Response: We thank the reviewer for the comments. The authors tried to match the concepts discoursed in the introduction section of the article with the scope the methodology, the results and discussion. Hence we mad the introduction brief and clear.
Comment 2: Conceptual background & Theoretical development:
This section needs to be established to clearly position the authors’ current study. There needs to be a clear link to extant studies (please see reference list for some useful suggestions) to ground the current study. The authors need to establish this section (or present a much longer introduction section which clearly links their research to extant studies) some clearer links to extant PPP literature (e.g. please see suggested key references) to guide the reader. This would help to clearly establish the basis of this paper. The author(s) should clearly draw out the benefits and limitations of PPPs and issues around the management of such long-term relationships to drive social responsibility and economic impact. This should then be linked to discussions around value for money, risks/uncertainties and sustainable development as key drivers. A much clearer positioning of the current study will help to further guide the reader and draw out gap(s) in extant studies.
Response: we acknowledge the reviewer’s comment and tried to link the suggested references with our work, but the authors believe that taking in account the methodology used and the research questions it can answer we believe having a longer introduction section could go out of scope.
Comments 3: Discussions and Conclusions:
This still needs substantial improvements. Derived from a conceptual background section which did not clearly draw out the gaps the paper seeks to address, the discussion (missing) and conclusion sections do offer no additional value to the reader as it stands. The authors need to offer more fine-grained results here and discuss what they intended to find out in the introduction section (link to research question; overall aim of the paper). Overall, the authors need to clearly draw out what the theoretical contributions are and how they add to the existing body of knowledge. This section also needs to clear link back to extant studies to offer some clear value to the reader.
Response: The authors tried to modify the discussion and concussion of the article according to the concepts and research gaps stated in the introduction section of the paper. We have included most of the references suggested by the reviewer.